# Microsite Drivers of Natural Seed Regeneration of *Eucalyptus globulus* Labill. in Burnt Plantations †

**Ana Águas** [1,2,*], **Hugo Matias** [1], **Abel Rodrigues** [3,4], **Tanya Bailey** [5], **Joaquim Silva** [1,6] and **Francisco Rego** [1]

1 Centro de Ecologia Aplicada "Professor Baeta Neves" (CEABN), InBio, Instituto Superior de Agronomia, Universidade de Lisboa, Tapada da Ajuda, 1349-017 Lisboa, Portugal; hgmg.matias@gmail.com (H.M.); jss@esac.pt (J.S.); frego@isa.ulisboa.pt (F.R.)
2 School of Education and Social Sciences, Polytechnic Institute of Leiria, 2411-901 Leiria, Portugal
3 Instituto Nacional de Investigação Agrária e Veterinária, Quinta do Marquês, 2780-159 Oeiras, Portugal; abel.rodrigues@iniav.pt
4 GeoBiotec, Faculdade de Ciências e Tecnologia, Universidade Nova de Lisboa, 2829-516 Caparica, Portugal
5 School of Natural Sciences, University of Tasmania, Hobart, TAS 7001, Australia; tanya.bailey@utas.edu.au
6 College of Agriculture, Polytechnic Institute of Coimbra, 3040-316 Coimbra, Portugal
* Correspondence: ana_aguas@yahoo.com; Tel.: +351-213653333
† This manuscript is part of a Ph.D. thesis by the first author, Ana Águas, and is available online at the link: https://www.repository.utl.pt/handle/10400.5/18326.

**Abstract:** Fire regimes are changing in several regions of the world. In those regions, some exotic species may be better adapted to new regimes than the native species. This study focused on identifying the microsite characteristics associated with the occurrence of post-fire *Eucalyptus globulus* regeneration from seeds, outside the species native-range. This information is important in helping to assess the naturalization status of the species, to understand its invasion risk, and to manage wildlings in plantations. To characterize the establishment niche, pairs of microsites (sapling presence/absence) were sampled in four salvage-logged plantations of *E. globulus* two years after fire (20 pairs/plantation). Microsites of wildlings from three size classes and control microsites were established in one of these plantations (20 quartets) in order to characterize the recruitment niche and to assess ontogenic niche shifts. Two post-fire wildling cohorts were identified. The first emerged just after fire and was abundant. The second emerged after logging and was scarce, probably due to seed limitation. First-cohort wildlings were observed in microsites characterized by a high incidence of fire-related variables (charcoal, ash, increased soil pH and K). The aggregated distribution of these wildlings and their association with other species may indicate the existence of facilitative relationships and/or the exploitation of resource-rich patches. All these factors were relevant for first-cohort persistence and likely also for its establishment and recruitment. Second-cohort wildlings occurred in microsites where salvage-logging disturbance was evident, showing the importance of this disturbance for its emergence. Wildling size diversity was explained by the two recruitment events and by the asymmetrical competition between wildlings and adults. No niche shifts were detected. The high densities of *E. globulus* wildlings found established in burnt plantations indicated naturalization was in progress. The timing of major recruitment events and the phenology of the species should be considered for monitoring this regeneration and scheduling control interventions, if required.

**Keywords:** *Eucalyptus globulus*; seed; regeneration; microsite; fire; ecological niche

## 1. Introduction

Many tree species are cultivated outside their native ranges worldwide. Fire is a factor that may affect the success of these species in their new environments [1,2], and this influence is heterogeneous both over time and space [3–5]. Furthermore, fire regimes are changing in many regions of the world, mostly as a result of changes in climate and land

use [6,7]. In such contexts, some exotic species are better adapted to new fire regimes than the native ones [1,2]. Several studies have been undertaken on cultivated trees outside their native ranges, for instance with *Eucalyptus globulus* [8,9], on several scales down to tens of square metres. However, environmental factors acting at smaller scales are particularly relevant for the recruitment and early survival of plants [10,11]. Although the microsite scale has not received as much attention as other scales, the knowledge on microsites of young plants in burnt plantations is important for plantation management.

The ecological niche is a keystone for understanding species distribution [12–14], as it encompasses the whole environment that allows species to exist indefinitely [12]. However, species have different ontogenic phases, which do not necessarily have the same requirements [15,16]. Indeed, it is proven that some species undergo ontogenic niche shifts [17–20]. Thus, the ecological niche is sometimes divided, for study purposes. For instance, the regeneration niche includes only the environmental factors that enable species to replace mature individuals by mature descendants [15]. Both reproduction and early ontogenic development are very sensitive stages for species perpetuation [15,21]. The availability of both seeds and safe sites for germination, *sensu* Fowler [22], is important for plant recruitment [23,24]. However, recruitment does not guarantee the species perpetuation because seedling mortality can be very high [25–27]. Successful establishment underpins that perpetuation. Complementarily to the concept of 'regeneration niche', Bond and Midgley [28] suggest the use of the term 'persistence niche' to encompass the factors that affect the permanence of established plants in situ. Persistence is achieved through the survival and growth of juveniles and adults, and it is determined by resource availability, temperature, early protection from stresses, litter accumulation, disturbance, competition, predation, symbioses, and pathogens [24,29–31]. Therefore, knowledge about the early life stages of the ecological niche, including establishment, is crucial when keeping plant populations under control.

Fire is an ecological factor that can influence plant distribution. The genus *Eucalyptus* includes many species adapted to fire [32]. This genus is native to Australia and a few islands in the Indian Ocean [33], and fire has influenced its evolution [34]. Fire is still frequent in many *Eucalyptus* forests, promoting regeneration from seeds, especially when intense [35,36]. It can affect *Eucalyptus* seed supply, as well as the availability and quality of microsites [24,37]. A short time after fire, *Eucalyptus* trees may release massive amounts of canopy-stored seeds [37–39]. Burnt sites are usually much more suitable for *Eucalyptus* seedlings as compared to unburnt sites [25,27,36,40,41]. Hence, fire may open a window of opportunity for regeneration from seeds, which lasts as long as the niche is still vacant [15,20]. Some authors go further with this idea. For instance, Mount [42] states that fire is required for *Eucalyptus* regeneration, in order to ensure stands' persistence, and Kirkpatrick [43] deems that a fire regime deeply influences the perpetuation of *E. globulus* in situ. Considering that many *Eucalyptus* forests are quite flammable [36,44–46], the paradoxical idea of flammability being an adaptation to fire [47,48] is probably true for many *Eucalyptus* species. Other authors [41] also postulate that increased seedling recruitment in burnt areas, together with the fire proneness of *Eucalyptus*, may generate positive feedback, favouring the colonization of new areas. Therefore, fire is a strong determinant of *Eucalyptus* distribution.

The fire influence on plant distribution is not spatially uniform. Fire behaviour depends on weather, topography, and fuel (type and distribution) [3,49]. Consequently, different microsite types are generated by fire within each burnt area [3,4], giving plants different chances to succeed. As a consequence, the regeneration success of *Eucalyptus* is also heterogeneous within burnt areas, being more frequent in sheltered microsites, with fire residues, moderately water-repellent soft soil, and low competition [20]. However, only a few studies have addressed the consequences of fire on *Eucalyptus* regeneration at the microsite level [20,24,50]; others focused on the microsite level but did not assess fire effects [17,51,52].

*Eucalyptus* is the most cultivated hardwood genus worldwide ($2.0 \times 10^7$ ha) [53]. Its cultivation is often controversial. On the one hand, it brings substantial economic benefits to land owners and industry [54,55]. On the other, it often has negative environmental impacts, namely decrease in biodiversity, invasion to contiguous ecosystems (rare and limited), decrease of water and nutrient availability in soil, and increase of fire hazard [54–56]. Nevertheless, most of these impacts may be prevented or substantially minimized by proper forest management [54–58]. Meanwhile, from the perspective of the exotic species, cultivation protects juvenile individuals against environmental stochasticity and preserves founder populations; thus, it may facilitate their naturalization and invasion in the arrival territories [59–61]. Moreover, synergies may happen between the wide recurrent cultivation of *Eucalyptus*, outside its native range, and the flammability of its forests, especially the poorly managed or abandoned ones [46,56,62,63]. As a result, large and intense fires may create plenty of good sites for *Eucalyptus* natural recruitment [58]. The success of eucalypts may be increased even further if fire occurrence and high propagule pressure coincide with good environmental conditions for regeneration from seeds [27,41,58,64]. However, apart from studies on *Pinus* spp., only a few others [1,41] have analysed the effects of fire on the reproductive success of trees cultivated in fire-prone areas, especially those outside their native ranges.

*Eucalyptus globulus* Labill. is native to SE Australia and Tasmania [43]. This species is perhaps the most widely planted of its genus worldwide. Its area was estimated to be $2.3 \times 10^6$ ha at the global level (data sources from 1977–2004 [65]), and it has been substantially expanded since then [66–68]. In Europe, this is the second most cultivated tree species, covering $1.46 \times 10^6$ ha, which is a much larger area than occupied by all the other eucalypts ($8.00 \times 10^4$ ha) [69]. Not surprisingly, *E. globulus* became naturalized in several regions of the world and is invasive in some of them (e.g., North America, Pacific Islands, and Europe) [70]. However, *E. globulus* has a modest invasive capacity, like other eucalypts, which have no clear ecological syndrome for invasiveness [56,71,72]. The vast majority of the descendants are closer than 15 m from the mother trees and are extremely rare at distances of 75–80 m [73–75]. While native forests and intensively managed stands restrain invasions by this species [54,58,64,75], natural drainage lines, dominant winds, and wildfires can facilitate them [41,74,76]. Notably, many stands of this species are in fire-prone regions [77–79], including the Western Iberian Peninsula, where they occupy the largest area outside the native range ($8.12 \times 10^5$ ha in Portugal; $1.74 \times 10^5$ ha in Galicia, Spain) [67,68]. The substantial expansion of the *E. globulus* afforested area had no effect on the Portuguese fire regime in the past four decades [80]. Nonetheless, fire often affects stands of this species in Portugal, and the fire hazard depends on stand structure and management [77,81]. Moreover, post-fire *E. globulus* regeneration from seeds is frequent, spatially heterogeneous, and sometimes attains high densities, especially in unmanaged areas there [58,64,82]. This species is often involved in post-fire land-use transitions related to land abandonment, tending to increase its representativeness in that territory [63]. Moreover, the increasing number of undermanaged and abandoned *E. globulus* stands, with high loads of continuous fuel, might influence the fire regime in the future [80]. Nevertheless, the interest in fire effects on regeneration from seeds of this species is fairly recent. The first studies were in the laboratory [83,84], and field-based research occurred only in the last decade [41,58,64,82]. Past investigations highlight the importance of interactions between fire and forest management for wildling recruitment and survival, but none were focused on the microsite level. Moreover, the spatial heterogeneity of this regeneration at the plantation level, which was formerly reported [8,9,85], is still largely unexplained. Studies at the microsite level would help to better understand this heterogeneous distribution and would provide useful information for forest management at a fine scale. Furthermore, studies in the introduction range of *E. globulus* would shed light on the naturalization status and the invasion risk of this species there.

The aims of this study were (1) to identify the microsite factors that are associated with the occurrence of *E. globulus* regeneration from seeds in burnt plantations; and (2) to

investigate the effects of factors related to fire and logging on the recruitment, establishment, and growth of this regeneration.

Henceforth, we will refer to the post-fire natural regeneration of *E. globulus* from seeds as post-fire regeneration. Similarly, the plants that comprised this regeneration will be represented by names referring to their condition of spontaneous plants (wildlings) or their ontogenic stage (seedlings or saplings).

## 2. Material and Methods

### 2.1. Study Sites

The study sites were four salvage-logged industrial plantations of *E. globulus* (Table 1). Two of them were located in central Portugal (Casal do Malta I, CM1; Casal do Malta II, CM2) and two in northern Portugal (Santo António, SA; Currelos Valdeias, CV). Sampling occurred in August–September 2014.

All the sites have a temperate climate with a dry and mild summer (Csb), according to the Köppen–Geiger classification [86]. However, important differences exist between the climatological standard norms of the closest weather stations (central Portugal: Dois Portos; northern Portugal: Nelas; 1961–90 period) (downloaded from http://home.isa.utl.pt/~joaopalma/tools/nc6090/ on 12 September 2015). As compared to the northern region, the central region has much lower annual rainfall (650.1 vs. 1007.8 mm), higher mean annual temperature (15.08 vs. 13.68 °C), less frost days (23.2 vs. 39.9 days year$^{-1}$), and weaker temperature seasonality.

**Table 1.** Characteristics of the studied sites (Casal do Malta I—CM1, Casal do Malta II—CM2, Currelos Valdeias—CV, Santo António—SA).

| | CM1 | CM2 | CV | SA |
|---|---|---|---|---|
| Coordinates (centroid) | 39°07′36″ N 9°13′06″ W | 39°07′53″ N 9°12′55″ W | 40°23′59″ N 8°00′25″ W | 40°18′05″ N 7°57′28″ W |
| Altitude | 75–100 m | 50–74 m | 176–226 m | 249–338 m |
| Lithology | Paleogene sandstone | Paleogene sandstone | Granite | (Pre-)cambrian schist-greywacke |
| Soil type | Eutric cambisol | Eutric cambisol | Humic and dystric cambisols | Humic cambisol |
| Mean annual rainfall (mm, period 1961–1990) | 650.1 | 650.1 | 1007.8 | 1007.8 |
| Tree density (trees ha$^{-1}$) | 1263 | 1041 | 788 | 840 |
| Rotation | 2nd | 2nd | 2nd | 2nd |
| Pole age at fire date | 10 years | 8 years | 6 years | 13 years |
| Burnt area | 14.90 ha | 10.59 ha | 7.35 ha | 14.00 ha |
| Wildfire date | 3 Sep. 2012 | 3 Sep. 2012 | 5 Sep. 2012 | 15 Sep. 2012 |
| Tree harvesting date | May 2014 | May 2014 | Jan.–Feb. 2013 | Nov. 2013 |
| Sampling date | Aug.–Sep. 2014 | Sep. 2014 | Aug. 2014 | Aug. 2014 |

Note: Lithological data from Silva [87]. Edaphic data from Cardoso et al. [88]. Rainfall data of the closest weather stations (Dois Portos, close to CM1 and CM2, and Nelas, close to CV and SA); downloaded from http://home.isa.utl.pt/~joaopalma/tools/nc6090/ on 12 September 2015.

The sampled plantations shared several attributes. They were even-aged monospecific plantations, exploited in a coppice system with rotations of 10–12 years. They were burnt in September 2012, approximately two years before sampling. At the time of fire, the studied plantations were at their second rotation, and their poles were reproductively adult. Afterwards, they were coppiced. Tree harvest consisted of a salvage logging performed by heavy machinery (harvesters and fellers), according to the standard procedures used in

industrial plantations. The poles of all planted trees were cut and removed from the sites. Although the remaining vegetation was not the harvesting target, it suffered side effects, for instance, many plants were injured while others were destroyed.

### 2.2. Sampling Design

The four plantations were surveyed for *E. globulus* regeneration from seeds. There, wildling sizes were in a continuum, within a wide size range. To study different developmental stages, target plants were chosen from clearly different size classes, representative of the minimum, intermediate, and maximum sizes (Table 2). These classes were named as seedlings (Se), short saplings (SSa), and tall saplings (TSa). Seedlings were considered as non-established plants, while saplings were considered as established, for three reasons: (1) mortality of *Eucalyptus* wildlings occurs mostly during their first summer [27,75]; (2) lignotubers enable eucalypts to survive severe stem damage [30,89]; and (3) *Eucalyptus* establishment takes at least 12 months [90].

**Table 2.** Characteristics of target plants, according to size classes.

|  | Seedlings (Se) | Short Saplings (SSa) | Tall Saplings (TSa) |
| --- | --- | --- | --- |
| Height (cm) | <18 | 30–100 | >150 |
| Stem architecture | single stem, ≤12 nodes | Branched | branched |
| Lignotuber width (mm) | 0 (absent) | ≤23 | ≥29 |
| Damage on stem | absent | present | present |
| No. of growth seasons of stem | 1 | >1 | >1 |

Twenty blocks of microsites were established within each burnt plantation, to study the post-fire regeneration. The CM1 plantation was chosen to study microsites of wildlings in different developmental stages, as it had the variety of wildling sizes best portrayed. Therefore, each block included one plot for each of the following four microsite types in CM1: microsites with no wildlings (C) and microsites with at least one wildling of a specific size class (Se, SSa, or TSa). In CM2, CV, and SA plantations, each of the 20 blocks included only one control plot and one SSa plot, to study wildling establishment. The distances between microsite plots within any block were always shorter (10–20 m) than between different blocks (>30 m). The plots were also 30 m or more away from the borders of the burnt areas. Apart from the aforementioned restrictions, the blocks were randomly placed, using a list of random numbers to choose the coordinates for where to start the search for each microsite type.

In every block, one plot was established for sampling each microsite type. The plots were circular and had a 25 cm radius. The control plots had their centre at a place with no *E. globulus* wildlings closer than 5.5 m. All the other plots were centred on a target plant (TSa, SSa, or Se), which was randomly selected amongst the available wildlings of the respective size class.

### 2.3. Field Sampling

In order to characterise each microsite, several attributes were assessed in each plot. Microtopography was characterized by aspect, slope, and microtopographical position of the central point (flat, slope, top, and depression). The cover of different vegetation guilds and ground cover types were visually estimated. The considered vegetation guilds were fungi, mosses, grasses, herbs, ferns, small shrubs (h < 1 m), tall shrubs (1 ≤ h < 5 m), and trees. Ground cover was classified as bare soil, rocks (diameter ≥ 2 cm), ash, charcoal, litter (leaves and twigs), and coarse woody debris (diameter ≥ 5 cm). The cover of each layer was independently estimated, using the following cover classes: **0**—absent; **1**—<1%; **2**—1%–4%; **3**—5%–10%; **4**—11%–25%; **5**—26%–33%; **6**—34%–50%; **7**—51%–66%; **8**—67%–75%; **9**—76%–90%; and **10**—91%–100%. The distance between the plot centre and the nearest vascular plant was measured. Conspicuous objects surrounding the plot centre (distance ≤ 2 m) were characterized according to height, width (degrees, measured

from the plot centre), distance from the plot centre, and type (rock, balk/talus, deadwood, living stump, shrub, or cluster of small plants). The presence of *E. globulus* capsules was also checked inside every plot. The depth of the mulch produced after fire was measured within 10 cm from the plot centre (5 readings), as were the ash depth (5 readings) and the soil hardness (10 readings with penetrometer, Geotester, Italy). In addition, a composite sample of surface soil was collected, using a soil borer (diameter = 2 cm; depth $\leq$ 5 cm; 5 cores). Finally, the maximum and median heights of all *E. globulus* wildlings around target plants were measured, and wildlings were counted within 1.78 m from plot centre (area = 10 m$^2$).

### 2.4. Laboratory Analyses

2.4.1. Soil Hydrophobicity

Soil samples were sieved (mesh = 2 mm), and a 15 g subsample of each one was placed into an individual Petri dish and left to air-dry until reaching a constant weight. Then, the subsamples were used to assess the persistence of hydrophobicity, using the 'water drop penetration time test' (WDPT), and the severity of hydrophobicity, using the 'molarity of an ethanol drop test' (MED) [91,92]. Air temperature and moisture were controlled during soil drying and hydrophobicity tests [93]. Temperature was kept at 19 $\pm$ 1 °C and relative air moisture at 50 $\pm$ 2%. Soil samples from the same block were analysed in a single batch in tests.

The WDPT test consisted of carefully dripping 5 drops of distilled water on the smoothed surface of each sample and measuring the median time to complete absorption (t) [91]. Whenever t > 1′, the Petri dishes were covered with a lid to reduce water evaporation. If t > 3 h, it was registered as such. The t of each sample was used to identify the respective hydrophobicity persistence class (Table 3).

**Table 3.** Specifications of soil hydrophobicity: (**a**) persistence, measured by water drop penetration time test (WDPT, adapted from Doerr [91] and Dekker and Ritsema [93]); (**b**) severity, measured by molarity of ethanol droplet test (MED) [94].

| | | | | | | | | | | | | | |
|---|---|---|---|---|---|---|---|---|---|---|---|---|---|
| **(a) Persistence of Soil Hydrophobicity** | | | | | | | | | | | | | |
| Water drop penetration time (s) | <5 | 5–9 | 10–29 | 30–59 | 60–179 | 180–299 | 300–599 | 600–899 | 900–3599 | 3600–7199 | 7200–10,799 | ≥10,800 | |
| Persistence class | 1 | 2 | 3 | 4 | 5 | 6 | 7 | 8 | 9 | 10 | 11 | 12 | |
| Descriptive category | Wettable | | Slight | | Strong | | | Severe | | Extreme | | | |
| **(b) Severity of Soil Hydrophobicity** | | | | | | | | | | | | | |
| Ethanol concentration (%, *v/v*) | 0 | | 1 | 2 | 3 | 5 | 8.5 | 13 | 18 | 24 | 36 | 50 | >50 |
| Severity class | 1 | | 2 | 3 | 4 | 5 | 6 | 7 | 8 | 9 | 10 | 11 | 12 |
| Descriptive category | Wettable | | Low | | | Moderate | | | Severe | | Extreme | | |

The MED test consisted in dripping 5 drops of an aqueous solution of ethanol on the smoothed soil surface and waiting for their absorption. Different solutions of decreasing concentrations were successively applied to each soil sample, until a single solution had t > 3″ [91,94]. The concentration of the last solution to be applied on each sample was used to identify the sample's class of hydrophobicity severity (Table 3) [94].

2.4.2. Soil Chemistry

Soil samples from all SSa and control plots were further analysed to determine texture, pH (in water), organic matter (%), and contents of extractable nutrients (P$_2$O$_5$, K$_2$O, and Mg$^{++}$; in mg kg$^{-1}$). As a preparation for the chemical analyses, all the samples were sieved (mesh = 2 mm) and oven dried (35–37 °C). Then, they were separated into subsamples for the different analyses. The texture was expeditiously ascertained by rubbing moist soil samples by hand (classes: coarse, medium, and fine). Soil pH was determined by potentiometry (Radiometer Analytical, France) in a water suspension (1:2.5, *v/v*; 1 h).

Colourimetry was used for quantifying organic matter (modified Tinsley method) [95,96]. Phosphorous and potassium were extracted from the soils with Egner-Riehm solution (soil/solution 1:20, $m/v$; 2 h), whereas magnesium was extracted with ammonium acetate at 1 M (soil/solution 1:10, $m/v$; pH 7; 30′). The concentrations of these nutrients were determined by inductively coupled plasma optical emission spectrometry (Perkin Elmer, USA) for $P_2O_5$; flame emission spectrometry (Corning, UK) for $K_2O$; and flame atomic absorption spectrophotometry (GBC Scientific Equipment, Australia) for $Mg^{++}$. As far as it was possible, soil subsamples from the same block were analysed in a single batch in every analysis.

### 2.5. Data Analyses

Exploratory analyses were made using the collected data from all plantations. These data included attributes observed in the field and in the laboratory, corresponding to control, Se, and SSa microsites. TSa microsites were excluded because we studied them only in CM1, despite their abundance in all sites. Covers of fungi, lichen, and rotten wood were discarded from the analyses because these guilds occurred in four or less plots. Exploratory analyses consisted of two steps. Firstly, a correlation matrix was built (Pearson $r^2$) using all measured attributes. Secondly, a forward stepwise discriminant analysis was performed. For this analysis, the grouping variable was 'plantation' and candidate-dependent variables were the 14 data attributes that had more than 10 significant correlations ($p < 0.05$) in the correlation matrix. Some variables were log-transformed (ln), in order to comply with the assumptions of discriminant analysis. Model quality was assessed through a classification matrix.

Exploratory analyses showed that the four plantations had substantially differed in their characteristics (cf. Equations (1)–(7) and Figure 1 in the Results' section). Hence, data from different plantations were separated for further analyses.

To compare SSa microsites with control microsites, the paired-sample Student's t test was used when data of continuous variables were normally distributed. The Wilcoxon test was used when those data distributions were not normal and for comparisons concerning ordinal variables. The latter test was used only when the data were symmetrically distributed around the median. When this assumption was not met, the Sign test was used instead [97]. McNemar's test was employed for comparisons concerning binary variables.

To compare the four types of microsites from CM1, a 2-way mixed-effects ANOVA (fixed factor—microsite type; random factor—blocks; no interaction between factors) and the post hoc Tukey test were used, when data had normal distributions. Alternatively, the Friedman test and the post hoc Dunn test were used when data distributions were not normal and for comparisons concerning ordinal variables. The Cochran's Q test and the Dunn test were chosen for comparisons concerning binary variables. Relationships between the microsite types and other nominal variables were analysed through contingency tables and the $\chi^2$ test [97].

Statistica 6.0 [98] was used for preliminary analyses, while SPSS 22.0.0.0 [99] was used for all the other analyses. Analysis results were considered significant at $p < 0.05$.

## 3. Results

### 3.1. Preliminary Observations

The planted *E. globulus* trees were the only individuals of this species that were damaged by fire in the studied plantations. Wildlings of different sizes, up to 3.9 m, abounded. Saplings were dominant in all plantations, whereas Se were a minority in CM1 and nearly absent in the other plantations.

### 3.2. Exploratory Analyses

Fourteen observed variables were significantly correlated to more than ten other observed variables ($p < 0.05$). They were southern aspect, cover of bare soil, cover of ash (cAsh), cover of small shrubs, soil hardness, soil texture, soil pH, content of organic

matter in soil, soil [$P_2O_5$], soil [$K_2O$] (soilK), soil [$Mg^{++}$], persistence of soil hydrophobicity (WDPT), number of wildlings, and median height of wildlings (medh).

Discriminant analysis for plantations produced a model that had an average Wilk's lambda of 0.36 and three discriminant functions with standardized coefficients (Equations (1)–(3)):

$$D1 = -0.60 \ln medh - 0.86 \ln soilK + 0.34\ WDPT + 0.39 \ln (cAsh + 1) \tag{1}$$

$$D2 = 0.73 \ln medh - 0.13 \ln soilK - 0.05\ WDPT + 0.60 \ln (cAsh + 1) \tag{2}$$

$$D3 = -0.22 \ln medh + 0.25 \ln soilK - 0.81\ WDPT + 0.54 \ln (cAsh + 1) \tag{3}$$

The classification functions of this model (Equations (4)–(7)) provided a classification matrix with 55 to 63.5% of correct classifications for the four plantations.

$$CM1 = 13.62 \ln medh + 32.33 \ln soilK - 0.78\ WDPT - 0.07 \ln (cAsh + 1) - 97.02 \tag{4}$$

$$CM2 = 15.31 \ln medh + 32.32 \ln soilK - 0.78\ WDPT - 0.07 \ln (cAsh + 1) - 103.94 \tag{5}$$

$$CV = 16.24 \ln medh + 37.06 \ln soilK - 1.16\ WDPT - 1.15 \ln (cAsh + 1) - 128.56 \tag{6}$$

$$SA = 16.50 \ln medh + 36.50 \ln soilK - 0.86\ WDPT - 2.26 \ln (cAsh + 1) - 127.64 \tag{7}$$

According to this model, the plantations can be distinguished using four variables (Figure 1; Equations (4)–(7)). Firstly, soils in CM1 and CM2 were poorer in potassium than those in SA and CV (Figure 1). Secondly, CM1 had the shortest wildlings comparatively with all the other plantations (Figure 1a), because it was the only plantation that had a significant number of Se. Thirdly, SA microsites were the least covered by ash, followed by those of CV (Figure 1). Finally, CV soil had the least persistent hydrophobicity (Figure 1b).

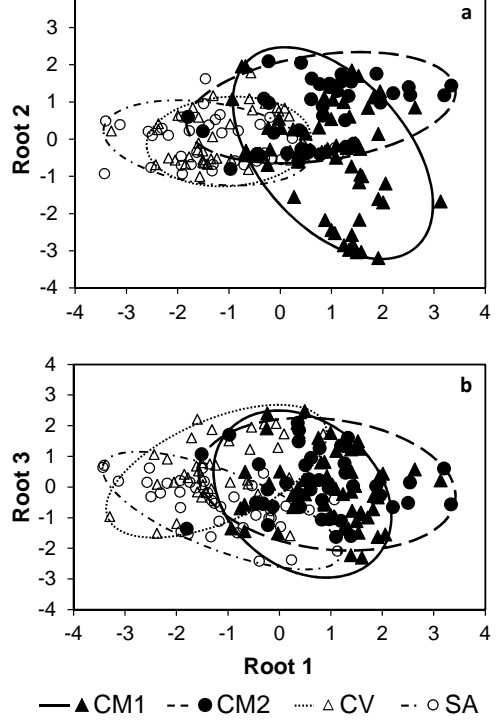

**Figure 1.** Scatter plots of canonical scores obtained in discriminant analysis of the plantations, using microsite attributes as dependent variables (see Equations (1)–(3) and text for detailed explanation): (**a**) root 1 vs. root 2; (**b**) root 1 vs. root 3. Plantations: CM1—Casal do Malta I; CM2—Casal do Malta II; CV—Currelos Valdeias; SA—Santo António (*n*: CM1 = 63; CM2 = 40; CV = 40; SA = 40).

### 3.3. Microsites of Wildlings of Different Sizes

Wildlings of various sizes were observed at all studied plantations. Their heights ranged from 3.5 to 390 cm. Comparisons of the respective microsites detected several significant differences in CM 1 (Figure 2 and Table 4). Firstly, when compared to TSa microsites, Se microsites were characterised by significantly shallower slope, less shelter, more bare ground, lower levels of soil hydrophobicity (severity and persistence), and less ash cover and depth (Figure 2a–j). These variables tended to have intermediate values in SSa microsites. Secondly, small shrubs covered substantially smaller areas in control and Se microsites than in sapling microsites, but post hoc tests were not able to determine where the differences were (cover class quartiles (Q1, Q2, Q3): control (0, 0, 1); Se (0, 0, 0); SSa (0, 1, 3); TSa (0, 2, 3)). Thirdly, the three types of wildling microsites significantly differed in terms of which types and how many other *E. globulus* individuals existed in the surroundings. Other wildlings occurred more frequently around TSa than around SSa and Se (wildling density quartiles, in plants m$^{-2}$ (Q1, Q2, Q3): Se (0.05, 0.40, 1.15); SSa (0.55, 1.00, 1.40); TSa (1.70, 2.80, 4.30)). Maximum wildling density observed was 9.90 plants m$^{-2}$, around TSa. Surrounding wildlings were also taller around TSa (mostly of class '50 < h ≤ 130 cm') than around SSa and Se (mostly of class 'h ≤ 50 cm'). Notably, the sharpest difference between microsites with SSa and TSa was the more than four-fold greater occurrence of live stumps close to SSa than to TSa (Figure 2k). Finally, the only difference between Se microsites and those without wildlings was the much more frequent presence of *E. globulus* capsules in the former (Figure 2l).

**Table 4.** Results of statistical analyses comparing different microsite types in Casal do Malta I plantation (*n* = 20). Microsite types: seedling, short sapling, tall sapling, control.

| Microsite Attribute | Test Statistic | *p* Value |
|---|---|---|
| Slope | $F_{3,56} = 5.313$ | 0.003 |
| Shelter amplitude | $\chi^2_r = 5.913$ | 0.001 |
| Shelter maximum height | $F_{3,56} = 16.207$ | <0.001 |
| Distance to nearest plant | $\chi^2_r = 10.600$ | 0.014 |
| Small-shrub cover | $\chi^2_r = 12.210$ | 0.007 |
| Bare soil | $\chi^2_r = 4.968$ | 0.002 |
| Charcoal cover | $\chi^2_r = 10.399$ | 0.015 |
| Ash cover | $\chi^2_r = 22.288$ | <0.001 |
| Ash depth | $\chi^2_r = 22.288$ | <0.001 |
| Persistence of soil hydrophobicity | $\chi^2_r = 25.532$ | <0.001 |
| Severity of soil hydrophobicity | $\chi^2_r = 21.475$ | <0.001 |
| Occurrence of alive stumps nearby | $Q = 16.833$ | 0.001 |
| Capsule occurrence | $Q = 12.488$ | 0.006 |
| Abundance of other wildlings | $\chi^2_r = 21.221$ | <0.001 |

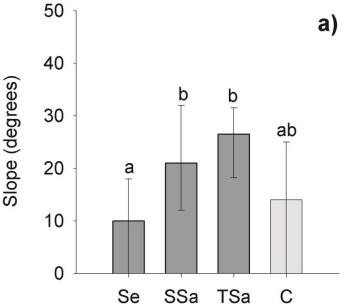 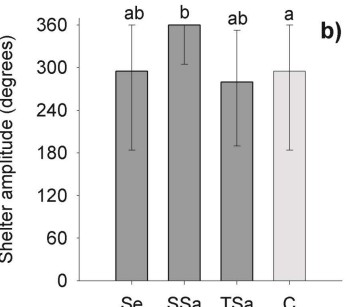 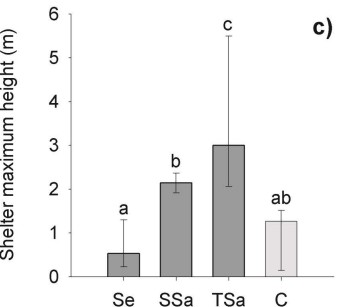

**Figure 2.** *Cont.*

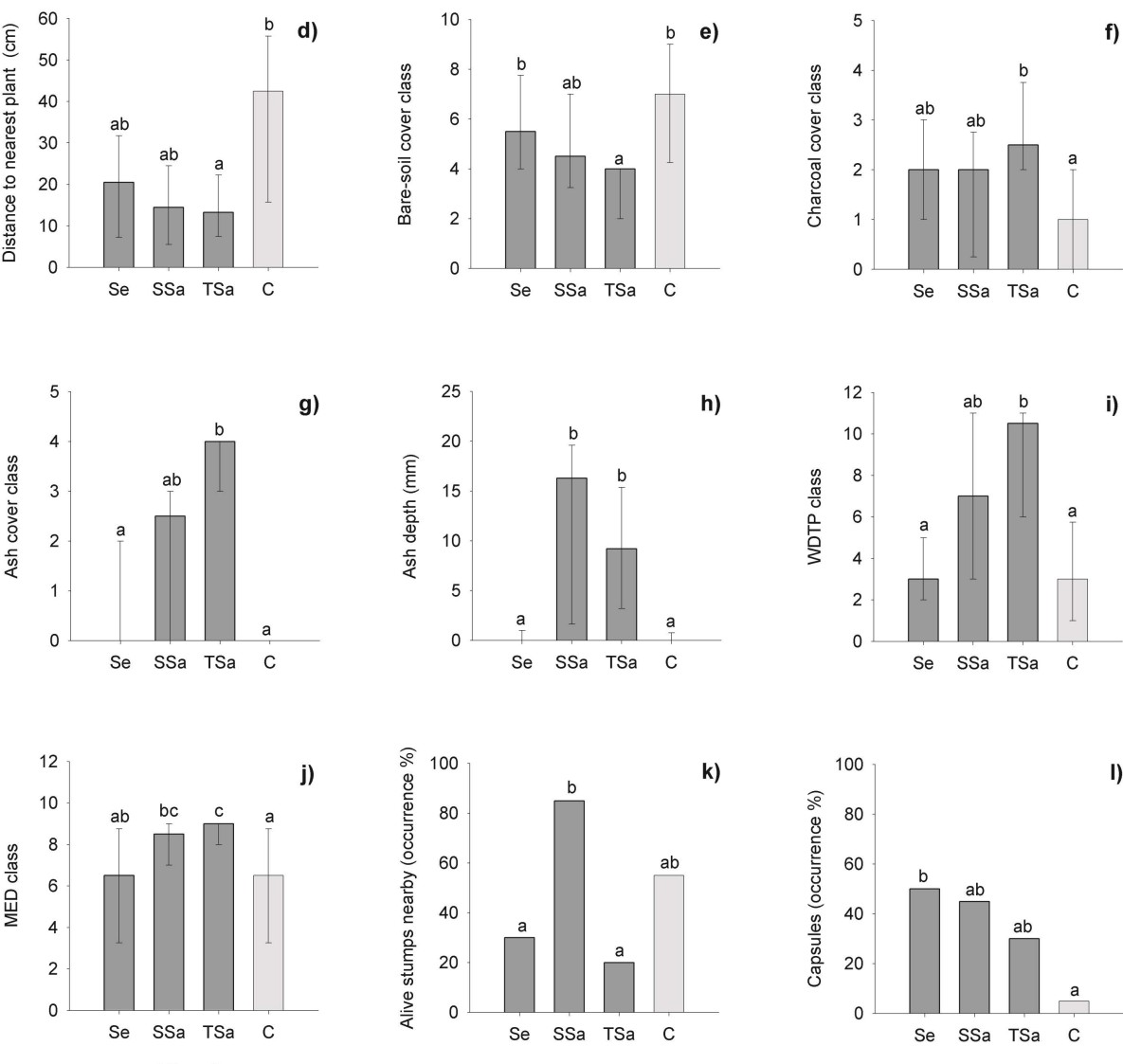

**Figure 2.** Microsite attributes (**a–l**) in the plantation Casal do Malta I (*n* = 20). Microsite types: Se—seedling; SSa—short sapling; TSa—tall sapling; C—control. Columns show medians in wildling microsites (▮) and control microsites (▯), and error bars show first and third quartiles. Microsite types that are significantly different (*p* < 0.05) are indicated by different letters. See text for detailed explanation about classes of cover, persistence of soil hydrophobicity (water drop penetration time test, WDPT), and severity of soil hydrophobicity (molarity of an ethanol droplet test, MED).

*3.4. Microsites of Established Wildlings*

Short saplings were regarded as the smallest established wildlings. Their microsites differed from those with no wildlings in several features (Figure 3), predominantly at the plantations of the central region (CM1 and CM2). While similar trends were found at the more northerly plantations (CV and SA), few of the differences (namely, slope, and ash depth at SA) were statistically significant (Figure 3). Microsites containing SSa, in comparison to those with no wildlings, tended to have: steeper slope; wider shelter; taller objects nearby; closer proximity to other plants: greater amount of small shrub cover; more charcoal cover; greater ash cover and depth; and soils with more persistent and severe hydrophobicity, higher pH, and greater concentrations of potassium (Figure 3). No other ground cover, plant cover, or soil properties differed significantly between SSa and control microsites.

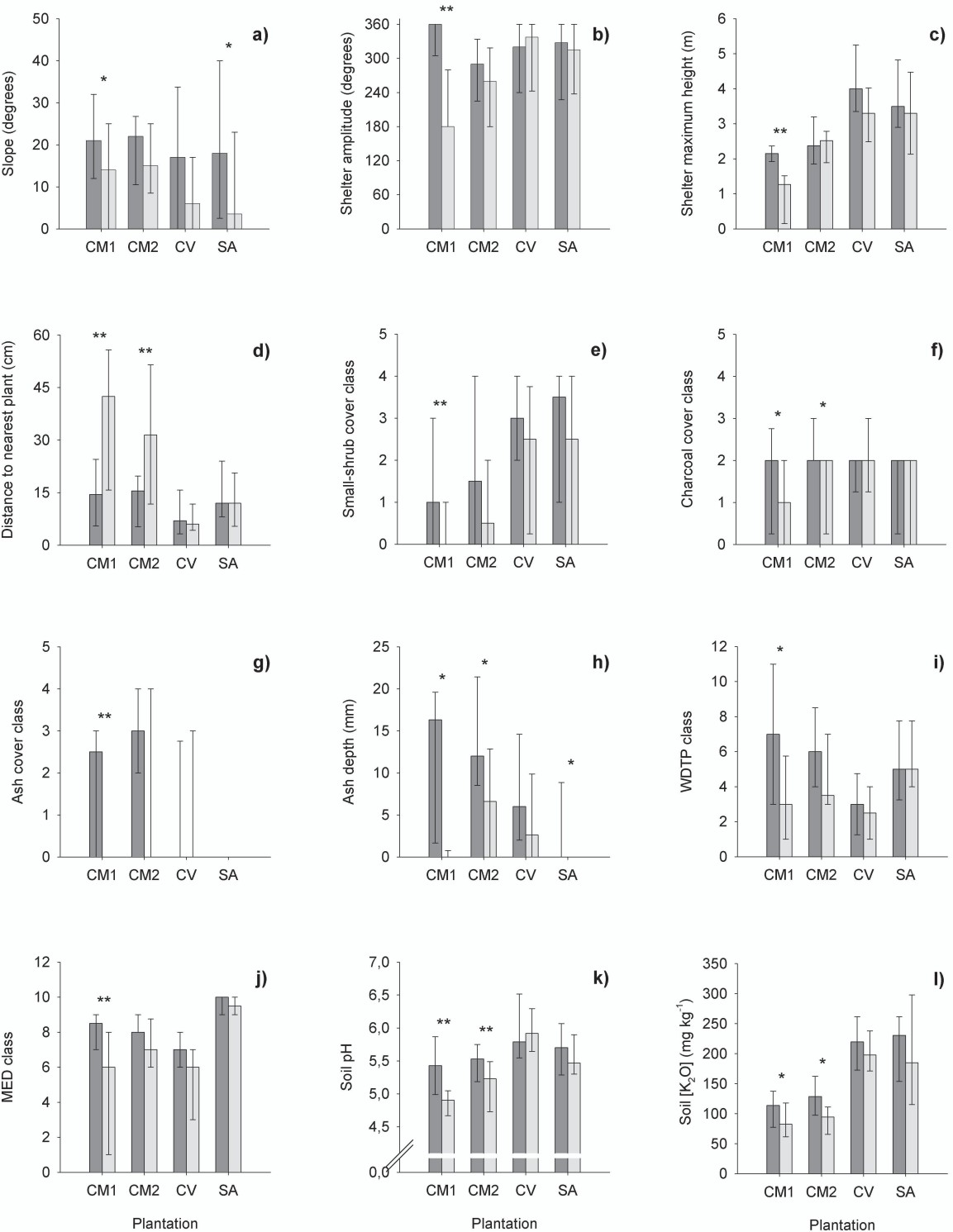

**Figure 3.** Microsite attributes (**a**–**l**) in the studied plantations. Microsite types: short sapling microsites
(▮) and control microsites (▯); (*n* = 20). Plantations: CM1—Casal do Malta I; CM2—Casal do Malta
II; CV—Currelos Valdeias; SA—Santo António. Columns show medians, and bars show first and
third quartiles. Significant differences are indicated by * 0.01 ≤ *p* < 0.05; ** 0.001 ≤ *p* < 0.01. See
text for detailed explanation about classes of cover, persistence of soil hydrophobicity (water drop
penetration time test, WDPT), and severity of soil hydrophobicity (molarity of an ethanol droplet
test, MED).

The height of objects surrounding the plot centres varied largely, even in a single plot, and ranged from a few centimetres to several metres. Each plot usually had several types of objects around its centre. No significant differences were found between SSa and control microsites, on the occurrence of each type of surrounding objects. Abiotic objects were represented (in 87.0% of the microsites) almost as much as biotic objects (in 93.8% of the microsites). The most common biotic objects around the plot centres were *E. globulus* stump resprouts (in 77.8% of the microsites) and plants of other species (in 50.6% of the microsites). Most of the abiotic objects were features derived from soil preparation for planting, namely 77.8% of the microsites had balks or taluses nearby. Distance from the plot centre to the closest object was not significantly different between SSa microsites and control microsites, in any plantation (mean = 28.1 cm).

## 4. Discussion

### 4.1. Origins of Wildlings

The studied plantations had thousands of wildlings inside burnt areas (range of measured density: 0–9.90 wildlings $m^{-2}$). As no artificial sowing was undertaken, these plants must have emerged from seeds of natural origin. Stems of planted trees (second rotation) were old enough to produce seed crops at the time of fire [43,100]. Seeds from these trees could easily reach any place within the plantations, considering the seed dispersal distance for this species [75,85,101,102], the height of planted trees when the fire occurred, and the plantation density (Table 1). This fact was confirmed by the detection of *E. globulus* capsules at similar frequencies at the sapling microsites and other microsites. The relevance of a pre-fire soil seed bank should have been negligible, because free eucalypt seeds usually do not last more than 6–12 months in the soil [36], and they are susceptible to heat [103]. However, many seeds could be stored and protected by capsules [39] when fire occurred (September 2012), and others could be produced by surviving trees later on. Therefore, the observed wildlings were the offspring of planted trees.

The observed wildlings ranged widely in size and had no fire injuries. Although *E. globulus* age is difficult to determine [104,105], the height of these wildlings is compatible with ages under two years [106]. Thus, wildling recruitment should have occurred after fire. Requisites for *Eucalyptus* regeneration are seed availability, mineral seedbed, suitable weather, and release from both predation and competition [25,32,35,36]. The conjunction of these conditions occurred twice recently in the sampled plantations. The first opportunity was just after fire, and the second was after salvage logging.

Fire destroys or damages competitors, destroys litter layer, and reduces populations of seed predators and browsers [107]. The heat-induced wilting of branches and the consequent drying of capsules enable seed release from *E. globulus* canopies a few weeks after fire [39]. When fire occurred in the studied plantations, trees could bear seeds in both capsules of the 2012/2013 crop [108] and serotinous capsules from previous crops. Seeds landing on soil in early autumn would have experienced favourable weather conditions for germination [109,110] and growth [111]. The resultant wildlings had the chance to grow for some months before winter frost and summer drought, thus improving their ability to endure those stressors [30,75]. This sequence of events and their timing must have enabled a numerous first cohort (observed saplings) to thrive in all plantations just after fire, as observed in other studies [58,64], which highlights the relevance of fire timing for wildling recruitment around seed bearing trees. Moreover, recruitment prior to salvage logging is likely due to both the estimated age of observed saplings (according to their height, signs of age, and lignotuber) and the physical damage detected on these plants, which could have been caused by tree harvesting.

An additional recruitment event seems to have occurred after salvage logging. Tree harvesting activities can elicit events of *Eucalyptus* regeneration, because they can expose mineral soil in some areas and provide an additional input of seeds [35,36,112,113]. In fact, logging left numerous patches of bare soil, and the harvested trees should have had a new seed crop that was ripe and ready for release since August 2013 [108]. Germinants could

emerge from both the viable seeds that already were on the ground at logging time and those released by the fallen trees [40,113]. Moreover, the sizes of observed seedlings were compatible with germination a short time after logging, and seedlings lacked ageing traits compatible with a second growth season. Hence, recruitment following salvage logging was the most likely origin of the second cohort (observed Se).

This second cohort was frequent only in one out of four plantations. Although fire was almost simultaneous in the four plantations, salvage logging was not. Differences in harvest timing determined not only the amounts of viable seeds that could land on the new safe microsites created by this operation, but also the weather conditions that those seeds and their germinants would have to cope with. Moreover, biotic interactions between *E. globulus* (seeds and Se) and other plants depended on both the phenology and spatial distribution of the coexisting species, which were different amongst plantations. Therefore, differences in second cohort density across plantations could have resulted not only from the tree harvesting timing but also from other factors that are out of the scope of this study.

Besides the three size classes analysed, plants of intermediate sizes did exist in the plantations. This fact indicates the possibility of continuous recruitment, as suggested by Silva et al. [64]. Seeds stored in fallen capsules might have been released later, when capsules were moved on the ground by the wind or landslides, or while capsule hoarders transported them to burrows (A. Águas, pers. obs.). However, the continuity of recruitment was certainly weak.

Considering the very high density of the first cohort, both seeds and safe microsites would have been very abundant just after fire and steadily decreased thereafter. Thus, the continuity of recruitment should have been restrained for a period. Later, salvage logging interfered with this trend. The microsites with Se were very similar to the controls, suggesting that the availability of safe microsites should not have been limiting for the recruitment of the second cohort (observed Se). Conversely, capsules were more often observed in Se microsites than in controls, suggesting that wildling absence in controls could be related to seed limitation [114]. Even after a supplementary input of seeds from slash, Se were much scarcer than saplings in the sampled population two years after fire. Therefore, the occurrence of a major recruitment event just after fire, followed by a period of much lower recruitment due to seed limitation, is the most likely explanation for the age structure of the observed populations.

### 4.2. Wildlings of Different Sizes

Differences in age and growth may explain the variety of wildling sizes in plantations two years following fire. Direct evidence of age are the common signs of ageing in perennial organs (e.g., secondary growth or scars) and the typical features of specific life stages (e.g., cotyledons, lignotuber, or branched stem). The observation of different combinations of these characteristics on target plants is not sufficient to determine their exact ages, but it clearly indicates that this regeneration from seeds is unevenly aged. The age differences have resulted from the major recruitment episodes and the possible continuous recruitment previously discussed.

Evidence of deep disturbance by tree harvesting in Se microsites included destroyed or damaged vegetation, exposed mineral soil, and altered soil profile (with reduced hydrophobicity [115]). While salvage logging has detrimental effects on some individuals and species, it opens an opportunity for others [116]. A substrate free from competition is a must for the survival of *Eucalyptus* seedlings [76]. They often grow up amongst unburnt slash, which protects them [113]. Therefore, salvage logging disturbance created a multitude of favourable microsites for Se survival and growth through reduced competition, some protection from physical stresses, and potentially increased water availability (by infiltration). Seasonality of climate can change some of these conditions to the point they become deadly to *E. globulus* seedlings. As these plants are very sensitive to drought, the first summer can be lethal to them, especially if they have germinated during the previous spring or in that summer [75,117]. However, the summer following salvage logging (2014)



was mild in central Portugal; misty mornings were common, and mean temperatures were 0.2–2.0 °C below the average of 1961–90 (data provided by Instituto Superior Técnico, Portugal). Therefore, the microsites created by salvage logging in spring turned out to be safe enough for Se during an extraordinarily mild summer. These results highlight the context dependence of wildling survival at the early stages of development.

In contrast to Se microsites, most attributes shared by TSa and SSa microsites were either direct or indirect consequences of fire occurrence (e.g., ash, plant clusters). This strongly suggests that TSa and SSa belong to the same cohort. The size differences between TSa and SSa may be explained by the way the microsites diverged in terms of the height of surrounding objects and the occurrence of conspecific plants nearby. Firstly, TSa had taller neighbouring objects than SSa. This extra protection from adverse conditions may have favoured TSa growth. Secondly, resprouting stumps were more common close to SSa than to TSa. These stumps were complete adult trees until several months before sampling, and they had a complete root system and strong resprouts at sampling time. Therefore, they were fierce competitors against the youngsters [50]. Competition largely suppresses growth rates of *Eucalyptus* seedlings and saplings beneath adult trees [118,119], and it can be extended to distances equivalent to 1–6 crown diameters [120–122]. Water and nutrients are the most likely resources under competition, between these saplings and adults [50,119,120]. Therefore, the occurrence of a very unbalanced competition between SSa and stumps is a sufficient cause for the size differences found between SSa and TSa, which presumably have a similar age. Thirdly, wildling density around TSa was three-fold higher than around SSa, and the wildlings surrounding each target plant belonged to its own size class. This suggests that these plants grow better away from adults, even if the alternative is to share space with a crowd of siblings [19]. Moreover, competition does not need to be weak for a biotic interaction to have a net positive outcome [123]. Our results indicate that the advantages of siblings living together overcame the disadvantages of competition among them, at least up to the TSa stage, even though *E. globulus* usually self-thins more rapidly than other eucalypts [36]. Additionally, tree harvesters did not venture into areas occupied by dense clusters of vegetation without mature trees to cut, and thus plants in those areas were safe from disturbance associated with harvesting. Comparable results can be found in studies focused on selective herbivory [19,124,125]. Nevertheless, the favourable balance of dense sibling-clusters would not last for long. The increasing demand for resources for growing individuals will increase competition, not only among siblings but also with adults. In fact, the height of saplings was already heterogeneous within clusters at sampling time, and the saplings may become stunted trees under the canopies of coppiced trees in the foreseeable future [36,82]. Therefore, even though continuous recruitment cannot be completely ruled out as a cause of sapling size diversity, the microsite characteristics indicate another scenario: saplings shared an origin connected to fire, and their size diversity was influenced by abiotic features and intraspecific relationships.

### 4.3. Wildling Establishment

The densities and sizes of saplings found in the studied plantations were comparable to those found at other burnt stands in Portugal, 6–7 years after fire [82]. All saplings were also within the size range of small reproductive wildlings of unknown age found in plantations in central Portugal [102]. In Australia, *E. globulus* individuals can start to produce flowers at 2 years old and fruits at 3 years [126]. Considering these facts, alongside the estimated age of saplings and the existence of lignotuber, the observed saplings are considered to be established and able to survive long enough to become reproductive.

Short saplings in CM1 occupied microsites that differed from vacant microsites in several features, including slope, shelter dimensions, proximity to other plants, small-shrub cover, ash and charcoal abundances, and soil traits. Consistent patterns were detected across plantations (SSa vs. control; Figure 3), despite the geographical and management differences (Table 1).

Slope was the only microtopographical feature that distinguished SSa from control microsites, being significantly steeper in the former than in the latter. These results are opposite to those observed with *Eucalyptus gunii* of a broader age range in a native forest setting [51], where water availability seems have limited wildling occurrence on steeper slopes. On the contrary, CV and SA had relatively high precipitation for *E. globulus* requirements; whilst CM1 and CM2 had bioindicator *taxa* of soil moisture (e.g., mosses, *Juncus* sp.), despite their region being close to the species lower precipitation limit [127,128]. Our results are in accordance with those obtained in study with three species of *Eucalyptus* of a similar age in SE Australia [52]. There, steeper slopes were associated with higher herbaceous cover, which may have allowed the trapping of seeds and resources, transported by superficial water flow. Ashton and Spalding [129] found that the vigour and density of eucalypt seedlings was patchy and related to the pattern of redistributed nutrient-rich hill-wash in burnt sites. This could be also the case in our study, since SSa plots tended to have higher plant cover, comparatively to the controls. Under stressful or low-productivity conditions, competition can be surpassed by facilitation as a determinant factor for seedling establishment [124,130,131]. Thus, the observed clusters of wildlings and shrubs might have created a synergy rather than just competition among plants, as observed by Fowler [22]. In fact, the nurse effect of shrubs is a common pattern in Mediterranean environments [19,124], and *E. globulus* wildlings are much more common in shrublands than in other habitats nearby plantations in Portugal [85]. Finally, steep slopes may also contribute to the occurrence of wildlings, through enabling small landslides that may bury seeds. Under a thin soil layer, seeds are fairly protected against predation, have more favourable conditions to germinate, and their radicles grow readily in contact with soil [36,132,133]. *Eucalyptus globulus* seeds germinate from depths down to 1.3 cm or 3.9 cm, depending on whether they are isolated or in groups [25]. Such depths are compatible with the occurrence of microtopographical landslides. Therefore, steep slopes may have improved the chances of sapling recruitment and establishment, by promoting seed burial and enabling facilitation phenomena.

The set of objects surrounding the plot centres tended to be taller and wider in SSa than in control microsites. These results can be related to the protective properties that those objects have in common, which can assist young plants to overcome climatic adversity [17].

Soils in SSa microsites tended to be less hydrophobic than controls. This result could be related to the plant clusters surrounding SSa. In fact, the severity of hydrophobicity was positively correlated with litter cover ($r^2 = 0.29$; $p < 0.001$; $n = 183$) and wildling height ($r^2 = 0.31$; $p = 0.002$; $n = 102$). This hydrophobicity is most likely due to the chemical composition of *E. globulus* litter (either under decomposition or burnt) and the high evapotranspiration of *E. globulus* trees, which dries soils during summer [134]. In fact, even hydrophilic soils become extremely hydrophobic 2 years after afforestation with *E. globulus* [135]. Although fire can induce or change soil hydrophobicity [136,137], it might have not been relevant in the studied plantations, as neither severity nor persistence was correlated with the other fire-related traits of microsites. Fire does not reinforce the strong soil hydrophobicity that exists before fire in *E. globulus* stands [135]. Therefore, the generally high soil hydrophobicity observed in our study plantations is likely to be both a legacy of hydrophobicity developed by the plantation before fire [94] and a byproduct of *E. globulus* plants after fire.

Combustion residues were the only ground cover features that differed significantly between SSa and control microsites, with SSa having more. The presence of ash beds (*sensu latu*: soils covered by either ash or charcoal) has been long considered a facilitator of *Eucalyptus* regeneration [25]. Pryor, in 1960, created the concept of 'ash-bed effect' [35,138] to name the enhanced plant growth achieved on soils that have been heated to temperatures usually in excess of 150 °C [139]. This stimulation of growth at early ontogenic stages increases the establishment chances of *Eucalyptus* individuals [35,129,140] and can explain the presence of SSa in places where fire was more severe.

Charcoal cover was wider on SSa soil than on controls in CM1 and CM2. Combustion can destroy phytotoxic substances [141], such as the phenolics contained in *E. globulus* leaves [142]. Charcoal itself can adsorb and deactivate phenolics in soil [143]. Their destruction or adsorption can have a positive effect on germination and plant growth [141,143]. The nitrifying microbial community is also affected by charcoal, increasing N availability to plants [144,145]. Charcoal can also adsorb nutrients, improving the cation exchange capacity of soil [146]. Furthermore, it can improve water retention and availability in coarse textured soils and, consequently, reduce nutrient leaching [146]. Thus, the strong presence of charcoal could have been influential in the emergence, growth, and survival of wildlings in CM1 and CM2, but it was not relevant in SA and CV, since the northern plantations had more favourable conditions for wildling development (higher rainfall and finer soils with more nutrients).

Ash tended to abound in SSa microsites in contrast to controls. Juveniles of several other *Eucalyptus* species were found to also have ash beds as preferential microsites [20]. Moreover, *E. globulus* shows competitive advantage over other eucalypts on ash beds [36]. This positive influence of ash on wildling development relies on physical and chemical mechanisms. Water retention in soil and water availability to plants can be improved when ash is produced by combustion above 300 °C and then is incorporated into the soil, because such temperatures break the hydrophobicity of organic matter [147]. The more complete the combustion is, the more alkaline and mineral-rich ash is [3]. Therefore, ash deposition on soils imposes important changes in soil properties, which favour plant development.

Soils tended to be less acidic in SSa microsites than in controls, but the differences were only significant in CM1 and CM2 plantations. Moreover, soil pH was positively correlated to ash cover ($r^2 = 0.24$; $p = 0.003$; $n = 147$). The soil pH increase may have been due to both ash deposition and soil heating [3,148]. Soil pH influences nutrient availability for plants: pH = 6.5 is generally considered as very favourable, while very acidic soils are frequently nutrient deficient (specially in P) and contain toxic levels of some other elements (e.g., Mn, Fe, Al) [149,150]. Phosphorous content and growth of ectomycorrhized *E. globulus* seedlings are larger when soils have pH 6 rather than pH 5 [151]. Acidic soils also reduce activity of nitrifying bacteria, reducing N availability for plants [149]. Our results match these ideas. Soil pH was positively correlated to soil K ($r^2 = 0.41$; $p < 0.001$; $n = 147$) and Mg ($r^2 = 0.77$; $p < 0.001$; $n = 147$). Importantly, many observed SSa were visibly ectomycorrhized. Additionally, CM1 and CM2 were the plantations with soil pH farther from the favourable range and with SSa present only in microsites with the least extreme soil pH. In the other plantations, soil pH did not seem to affect SSa distribution, because it was generally close to the favourable range. Therefore, as far as soil pH improves plant nutrition, it can be a factor that favours wildling establishment.

The soils of SSa microsites tended to be richer in K available to plants than those of control microsites, but differences were significant only in plantations with lower K concentrations and less rainfall (CM1 and CM2). Additionally, soil K was also positively correlated with ash cover ($r^2 = 0.23$; $p = 0.003$; $n = 155$). The ash of *Eucalyptus* spp. litter is dominated by Ca and K [3,148]. Hence, ash deposition increased the K in soil. This nutrient plays important roles in plants. It is one of the three most abundant mineral nutrients in the stems and leaves of *E. globulus* [152], and it is involved in phloem translocation, cell expansion, and protein synthesis [153]. Foliar K content of 1-year-old *E. globulus* has a strongly positive correlation with growth in the following years [154]. Thus, the abundance of this nutrient must have been a determinant for wildling development. In addition, K is important for plants to deal with stresses, enhancing their resistance to diseases and pests, as well as their tolerance to drought and frost [153]. Notably, these are the main climatic restrictions to *E. globulus* in Portugal [155]. This species has relatively high water potential thresholds for stomatal closure [26], and it is able to make osmotic adjustment [156], processes in which K plays a fundamental role [153]. Though the K concentrations in soil were medium for fertility standards [149], in the plantations with the poorest soils (CM1 and CM2), the higher concentrations in the SSa microsites may have helped wildlings

to grow and better cope with stressors. This may have been particularly useful to those SSa that were next to their major competitor, the adult trees, or their resprouting stumps (90.0% in those plantations). Hence, K contributions for SSa growth and water balance may have improved their survival chances in central Portugal. Conversely, K availability was not an issue in the northern region. There, water stress events were less likely, due to climate, and soils were generally very rich in K [149]. Thus, the SSa needs of K and its supply were more balanced.

In short, the presence of ash beds, sheltering objects, and microtopography have favoured the healthy development and survival of wildlings, enabling their establishment in burnt plantations. These results are particularly relevant, proving the resistance of wildlings and the role of the post-fire environment, which enable establishment even under severe competition.

### 4.4. Ecological Niche and Ontogeny

An overall comparison shows that microsites with wildlings of different sizes are distinguishable. Unoccupied microsites diverge from the microsites of the three wildling classes to different extents, being most similar to the Se microsites and least similar to TSa's, with SSa's being intermediate. These results could suggest a narrowing transition of niche that would gradually occur while wildlings grow old, as observed in *Acer opalus* [19]. However, most differences between sapling microsites and the other microsites are related to fire and salvage logging, two triggers of wildling recruitment. Therefore, the two cohorts had different recruitment niches. The first was strongly shaped by fire, and the second by logging. This fact prevents the determination of whether the observed wildlings had undergone an ontogenic niche shift or not.

Other studies [19,20] succeeded in the characterization of both recruitment and persistence niches and in the assessment of niche shifts, following a methodology that was similar to ours; however, they considered a longer timeframe. It is possible that observed wildlings have not undergone a niche shift as yet or that they never will, as the fire effects are still strong in the microsites of the saplings. Moreover, a major post-fire disturbance, salvage logging, made the dynamics of the studied populations more complex in our case. A follow-up sampling could solve some doubts raised by these results. However, the clear-cut separation of fire effects from logging effects on *E. globulus* regeneration requires further studies under experimental conditions and during a longer period after fire.

Nevertheless, salvage logging is a very common management operation in reproductive burnt plantations of *E. globulus*. Hence, the inclusion of this factor was an opportunity to obtain additional information about the post-fire microsite factors underlying the sexual regeneration of *E. globulus* in such plantations.

### 4.5. Implications in Forest Management

The observed wildling densities are very high, and they may affect subsequent management operations, consolidate the naturalization process, and increase both the invasion and fire risks. Specific management is required to control this regeneration, as suggested by several authors for other exotics species [157,158]. Considering the ontogeny and phenology of *E. globulus*, it is recommended to inspect the burnt plantations at the end of the first summer after fire, in order to evaluate the density of wildling cohort(s) and the need to control them. At that time, the alive wildlings are *ca.* 1 year old or less. They would just have proven their resistance to the major selective factor of summer drought [27,75] and would be about to start developing their lignotuber, which enables even greater resilience to adversities [30,89]. Simultaneously, they are still small enough to be easily destroyed by simple management operations [58,64].

Salvage logging can be an opportunity to either increase or reduce the number of wildlings in burnt plantations. As discussed before, logging usually keeps most of the first cohort alive, temporarily reduces competition, and creates conditions for the recruitment of a second cohort. However, the first cohort could be destroyed at the time of salvage logging

if forest managers decide to do so. This would be particularly successful if wildlings were *ca.* 1 year old or less, and it would be cheaper if the first cohort had already passed through a major natural selection event, reducing the number of wildlings to destroy. Additionally, the emergence of a second cohort can be reduced with the removal or burial (at depths > 5 cm [25]) of seed-bearing slash, but never with slash burn [113]. Most importantly, this second recruitment can be prevented if salvage logging occurs before burnt trees produce a new seed crop at up to 6–10 months after the first post-fire blooming [108].

## 5. Conclusions

Both fire and salvage logging create environmental conditions for the occurrence of *E. globulus* regeneration from seeds in burnt plantations. Each of these events may enable the emergence of a large cohort of wildlings, even in plantations without natural recruitment before fire.

The first cohort is numerous and is recruited a short time after fire. Fires occurring in late summer in *E. globulus* plantations create a synergy with the species phenology, enabling massive recruitment. Two years later, high densities of wildlings still exist. These plants have survived logging disturbance and are established. Fire effects on the physical and chemical environment clearly last for two years at least. They should be as important for wildling recruitment as they are for the establishment and persistence of those plants two years after fire. First-cohort wildlings mostly occur at microsites crowded with other plants, where resources are abundant and facilitative relationships seem to prevail over competitive ones. However, competition between wildlings and adult trees or resprouting stumps is strong, clearly hindering wildling growth, despite not preventing establishment. This asymmetrical competition is the major reason, besides age, for the size differences amongst wildlings.

The second cohort is recruited after salvage logging and is much scarcer than the first, probably due to seed limitation. This cohort inhabits microsites with exposed mineral soil resulting from salvage logging and reduced fire-related traits, showing that the recruitment conditions were substantially different from those experienced by the first cohort. Nevertheless, water availability, reduced asymmetrical competition, and shelter were as crucial for these plants as for the others.

Although tree harvesters usually allow wildlings to persist in plantations, salvage logging can be an opportunity to control the first cohort, if required, and to prevent the recruitment of the second. In this context, the timing of salvage logging is crucial.

In conclusion, the heterogeneity of *E. globulus* wildling distribution in stands can be explained by microsite diversity and seed availability. Fire and salvage logging at reproductive stands can induce seed release and create safe microsites. Therefore, attention should be paid to the effects that wildfires and salvage logging have in enabling and promoting *E. globulus* regeneration from seeds. The potential weed risk of this regeneration in areas outside the native range should be assessed and considered in forest management.

**Author Contributions:** Conceptualization: A.Á., J.S. and F.R.; methodology: A.Á., H.M., T.B., J.S. and F.R.; formal analysis: A.Á., A.R. and F.R.; investigation: A.Á. and H.M.; writing—original draft preparation: A.Á.; writing—review and editing: A.Á., H.M., A.R., T.B., J.S. and F.R.; supervision: A.Á., J.S. and F.R., funding acquisition: A.Á., J.S. and F.R. All authors have read and agreed to the published version of the manuscript.

**Funding:** This research was funded by Fundação para a Ciência e a Tecnologia (FCT, Portugal) through the project PTDC/AGR-FOR/2471/2012, and also through the research units UIDB/04035/2020 (GeoBioTec) and UIDB/50027/2020 (CEABN). A.Á. was supported by FCT [scholarship SFRH/BD/76899/2011] and by the European Commission [Marie Curie Actions, TRANZFOR programme].

**Institutional Review Board Statement:** Not applicable.

**Informed Consent Statement:** Not applicable.

**Data Availability Statement:** Not applicable.

**Acknowledgments:** Chemical analyses of soil were made at Laboratório Químico Agrícola Rebelo da Silva of Instituto Nacional de Investigação Agrária e Veterinária. Meteorological data from 2012/14 were provided by Instituto Superior Técnico of Universidade de Lisboa. We thank Altri-Florestal S.A. for providing access to their plantation estate and to data on plantation characteristics. Special thanks to Luís Ferreira for helping with Altri's database and to Luís Leal for reading and commenting on the manuscript.

**Conflicts of Interest:** The authors declare no conflict of interest. The sponsors had no role in the design, execution, interpretation, or writing of the study.

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
