# Peer review of "Microsite Drivers of Natural Seed Regeneration of Eucalyptus globulus Labill. in Burnt Plantations"

_forests, doi:10.3390/f13060889_

Round 1

Reviewer 1 Report

1. Are Eucalypt or Eucalyptus globulus at risk of invasion in Portugal or the Mediterranean region in southwestern Europe? Is it an invasive species? If so, it is suggested that the description of this aspect in Introduction (Eucalyptus is the most cultured hardware gene worldwide...) should be placed beginning the text.

2. As far as I know, eucalypt naturally regenerates very slowly, for example, compared with pine. At the same time, southwest Europe is rainy in summer and humid and mild in winter. From the perspective of competition for water resources, it should not be a big problem. Just like in South China, the debate about eucalypt has always existed, but the real problem is whether eucalypt? this tree? Short rotation period, high-density planting, one clone / tree species large-scale planting and other managements are the problem. Therefore, the authors should also discusses from this aspect, rather than simply talking about E. globulus.

3. Are “The high densities of E. globulus wildlings found established in burnt plantations indicate naturalization is in progress” and “No niche shifts were detected” contradictory?

4. It is suggested to supplement the rainfall information in Table 1.

Author Response

Dear Reviewer

We thank you for your comments. They allowed us to improve the manuscript.

Below we will answer to each comment, presenting and clarifying our ideas and stating the changes we have done in the manuscript. Your comments are shown in bold type and each of our answers is writen in roman type just belowthe respective comment.

0.  Extensive editing of English language and style required

The manuscript was thoroughly revised by two English native speakers.

1. Are Eucalypt or Eucalyptus globulus at risk of invasion in Portugal or the Mediterranean region in southwestern Europe? Is it an invasive species? If so, it is suggested that the description of this aspect in Introduction (Eucalyptus is the most cultured gene worldwide...) should be placed beginning the text.

Invasion is an issue that is clearly outside the scope of this study, despite being related with it. We studied recruitment inside plantations, where invasion conceptually is impossible. Our aim was to understand naturalization, analysing microsite driver of recruitment. All invasions are preceded by naturalizations, but only some naturalizations are followed by invasions. Eucalyptus spp., in general, and Eucalyptus globulus, in particular, is considered a mild invasive species in the Iberian Peninsula and some other regions of the world. Proper management, common in Iberian industrial plantations, substantially reduces its invasive capacity. Under this type of management, stems are coppiced at 10-12 years old, when they already are reproductive but are far from attaining their maximum propagule output. Other management operations, such as thinning and understorey removal, frequently performed in such plantations are opportunities to destroy wildlings. Wildlings surviving to those operations can be easily detected both inside and outside plantations, because of their typical habit and conspicuous colour (light blue). Moreover they can be controlled, by cutting or using herbicides. Invasion is more likely when plantations are not managed at all or following fire. Therefore, invasions by E. globulus usually are not a problem in the Iberian Peninsula, when stands are properly managed and fire does not occur. Under such circumstances, they are uncommon, easy to detect, and easy to control. The real problem may occur in unmanaged areas, following late summer wildfires. These areas deserve special attention (monitoring) and control methods exist for the possible invasions. Considering these ideas, we decided to address invasion a bit deeper in the introduction of the manuscript, but we considered this subject does not deserve a highlight. We expect this approach meets the reviewer’s request.

2. As far as I know, eucalypt naturally regenerates very slowly, for example, compared with pine. At the same time, southwest Europe is rainy in summer and humid and mild in winter. From the perspective of competition for water resources, it should not be a big problem. Just like in South China, the debate about eucalypt has always existed, but the real problem is whether eucalypt? this tree? Short rotation period, high-density planting, one clone / tree species large-scale planting and other managements are the problem. Therefore, the authors should also discusses from this aspect, rather than simply talking about E. globulus.

Once again, the controversies about the cultivation of eucalypts are outside the scope of our study. Considering the reviewer’s comment, we have decided to briefly mention them in the introduction, and referrer that most of the problems related to eucalypt cultivation can be kept under control (or even eliminated) with proper forest management. We only gave more details on the issues related to our study (invasiveness and fire) and we kept the focus on E. globulus, not only because it was our target species but also because it is the most cultivated in worldwide and specially in Europe.

Eucalypts are controversial species in Europe, as in many other parts of the world where they are widely cultivated. E. globulus is the most planted eucalypt in Europe and occupies a huge area as compared to the other eucalypts. Controversy has multiple sources. Land owners and the pulp and paper companies have economic interest on the cultivation of this species. However, the cultivation of this exotic species has ecological impact. Below wer present those of major concern:

  • Eucalyptus grow faster than many other cultivated or spontaneous tree species, consuming the corresponding amounts of water and nutrients from soil.
  • The very large areas this exotic genus occupies, in detriment to the native species (often confusing planted areas with invaded areas).
  • The monospecific stands have less biodiversity than (semi-)natural forests. When those stands are contiguous to other similar stands, vast areas have the very same problem which upscale to the landscape level.
  • Wildfires occur frequently in the afforested area where this species is dominant. This frequency is mainly due to the very large size of this area (~25% of the Portuguese forests). Moreover, the spread and intensity of fire are often aggravated by the land abandonment or poor management of fuels. Although eucalypts are known for some features that can increase flammability, stands of this species may range from little to large fire susceptibility, depending on the stand management and structure.

3. Are “The high densities of E. globulus wildlings found established in burnt plantations indicate naturalization is in progress” and “No niche shifts were detected” contradictory?

These sentences are not contradictory. Next, we are going to explain them in detail, to clarify the doubt expressed by the reviewer.

We have found high densities of wildlings in burnt plantations. The wildling abundance is relevant for the naturalization. This process happens when the population(s) of exotic species become able to persist in the arrival territory indefinitely, without human intervention. When planted exotic trees produce offspring without human support, the first step for naturalization is achieved. If this offspring is abundant, the chances to have individuals that survive to reach the adult stage are quite large. As most wildlings we have observed were at the sapling stage, they already have overcome the critical seedling stage, during which mortality rate is usually the highest of the whole ontogeny. The observed high densities of lignotuberous saplings are a clear sign that the production of a second generation of wildlings is very likely, since the observed wildings were 1-2 years away of becoming reproductive and they already had reached body sizes similar to other conspecific reproductive individuals observed central Portugal. Thus, evidence was observed that naturalization is occurring.

Ontogenetic niche shifts are changes which occur over the life cycle of species. We have not detected such shifts in the observed cohorts. For detecting an ontogenic niche shift in a scientific study two facts must be true: the niche shift must really occur in the studied species/population; and its assessment should be done with appropriate methods and at the right time. However, some species/populations simply do not undergo evident niche shifts. In addition, some others do undergo such shifts but they are undetected because the assessment was not properly done, either because it was not used the best method or it was not done at the right time. In our study, we have used a cross-sectional approach, sampling 2 cohorts of wildlings at the same for assessing the environmental conditions of two different life stages. This method was efficient at detecting niche shift in other studies. However, we have observed 2 cohorts emerged in quite different circumstances one form another, a fact which was unknown before sampling. The different contexts of emergence and development of these cohorts prevented us to use the data about the youngest cohort as a proxy to characterize the early stages of the oldest cohort. Thus, the collected data did not allow any conclusion about the existence of a niche shift between the seedling and sapling stages. To detect niche shifts we should have chosen another timing for sampling or we should have used a follow-up approach to observe each cohort separately, as we pointed out in section 4.4 (ecological niche and ontogeny).

If we had detected niche shifts in the observed populations, we could have identified factors enable young recruits to survive and go forward to close the life cycle. These would be probably factors that make the difference between mere existence of wildlings, resultant from recruitment, and their persistence in situ until a reproduction event, i.e. a key to naturalization.

The two sentences are not contradictory for two reasons: Firstly, we did not detect niches shifts but our results do not allow us to state that they did not exist. Secondly, naturalization may be related to ontogenic niche shifts, but this relationship in not mandatory.

4. It is suggested to supplement the rainfall information in Table 1.

The climatological data, including rainfall, had already been presented in the text. Following the reviewer’s suggestion, we inserted rainfall information in table 1, as well as the source of this information in the table’s footnote.

We hope you find the manuscript suitable for publishing, and we are available for any further work or clarification, if you find necessary.

Regards

The authors

Reviewer 2 Report

This is a good quality article. The aim is clearly defined and the study is well described.

The language is good and the structure of the article is logical and well-considered.

Authors have collected a wide range of actual articles on studied issue.

Especially the Discussion chapter of the article is worth to highlight as very comprehensive and good shaped.

Some information is missing about key characteristics of salvage-logged stands prior to fire as well the logging procedure, however, if such information was not available for researchers this is not big issue.

Author Response

Dear reviewer,

We thank you for your comments. They were very flattering and allowed us to improve the manuscript.

The manuscript was thoroughly revised by an English native speaker, to eliminate the linguistic mistakes and improve discourse style.

We also have added the required information. We have transcribed your request to this message (in bold type). Please see the answer to this request below. 

Comment "Some information is missing about key characteristics of salvage-logged stands prior to fire as well the logging procedure, however, if such information was not available for researchers this is not big issue.

Some information about the plantations and the tree harvesting is requested but it is not specified which information. We tried to accomplish the request, hoping to present the specific information you deemed as necessary.

  1. Key characteristics of stands prior to fire

The information concerning the plantation at fire moment, which is presented at table 1, is the usual and sufficient for characterizing the plantation. In Portugal, E. globulus is cultivated in even-aged monospecific plantations, exploited in a coppice system of 3-4 rotations of 10-12 years each. In the current revision, we added some information and highlighted other information previously presented, in order to facilitate reading.

In the original manuscript we have mentioned the studied plantations are industrial plantations and provided information about: rotation, age of poles, and tree density. During the revision we inserted the information about rotation in the table 1, so the reader can analyse that piece of information together with other pieces concerning the status of each plantation at the fire moment. In the text, we added brief description of general characteristics of those industrial plantations: even-aged monospecific plantations, exploited under coppice system with rotations of 10-12 years (see text of section 2.1., above table 1).

Considering the aims of this study, rotation, pole age, and tree density are key characteristics of the plantations. They inform the reader about the capacities the planted trees have to produce seeds and to sprinkle the whole ground surface with those seeds. These trees were at the reproductive age. Any of the planted trees was no more than 4 m away from another tree, a smaller length than their own height. Thus, the seed shadow of the canopies covers the entire soil surface within the plantations. This assumption is the crucial point to argue that all observed wildlings could be descendants of the planted trees.

Finally, we consider that it could be interesting for the reader to have the site index and the tree height for the 4 study sites. We have asked the land owners these data, to insert them in table 1. We are waiting for the information in the coming days. We would be happy to insert this information in table 1 as soon as we receive it, if the reviewer finds it necessary and the editor allow.

  1. Logging procedure

Tree harvest in the study sites consisted of a salvage logging, performed by heavy machinery (harvesters and fellers), according to the standard procedures used in industrial plantations. The poles of all planted trees were cut and removed from the sites. The remaining vegetation was not a target of the harvesting operation but suffered its side effects. Some spontaneous plants were destroyed and many were injured by the harvesting machinery and falling logs. During the current revision, we inserted this information in the text of section 2.1., above Table 1

We hope you find the manuscript suitable for publishing, and we are available for any further work or clarification, if you find necessary.

Regards

The authors